# When Matilda shows up: The double-edged impact of women researchers' media visibility in management science

Béatrice Parguel[1], Marine Agogué[2]*, Julie Gauneau[2]

1 CNRS, Université Paris Dauphine – PSL, Paris, France, 2 HEC Montréal, Montréal, Quebec, Canada

* marine.agogue@hec.ca

## Abstract

Although the proportion of women in academic positions has increased, women researchers continue to receive less media coverage than their counterparts. To examine the implications of this underrepresentation, we investigate how different audiences respond to women researchers' increased visibility in social sciences dissemination media. We conducted two laboratory experiments, the first with a sample of non-academic individuals (N = 271) and the second with a sample of graduate students (N = 129). Results show that greater visibility of women researchers reduces the gender gap in self-efficacy beliefs between men and women in both populations. It also increases women researchers' perceived expertise among non-academic participants, but not among graduate students. Among the later, however, heightened visibility is also unexpectedly associated with a decline in the perceived attractiveness of academic careers, but only for men. While our research focuses specifically on women researchers in the field of management science and manipulates media visibility through the relative representation of women and men in a simulated scientific dissemination newsletter, our findings offer broader insights into how media exposure can influence self-efficacy beliefs, perceptions of researchers' expertise, and the attractiveness of academic careers. Building on these insights, we propose recommendations for researchers, academic institutions, and media organizations engaged in science dissemination.

## 1. Introduction

As Oscar Wilde's Lord Henry remarks in *The picture of Dorian Gray*, "*there is only one thing in the world worse than being talked about, and that is not being talked about*" [1]. The same holds true across academic domains, where visibility functions as a form of social capital, built when others recognize a researcher's name, are aware of their work, and regard their intellectual contributions positively [2]. In the competitive world of "publish or perish", academic visibility often depends on

**Data availability statement:** All relevant data are within the paper and its Supporting Information files.

**Funding:** The author(s) received no specific funding for this work.

**Competing interests:** The authors have declared that no competing interests exist.

publication in high impact factor journals: across disciplines, the more academics publish, the more they get cited, and the more visible they are [3,4]. This academic visibility can be assessed via Web of Science or Google Scholar. But besides publications and citations, academic communities are increasingly turning to multidimensional "alternative metrics" [5]. These include academic visibility in the traditional [6,7] and social media (e.g., the number of times a paper has been tweeted, liked, downloaded, shared or cited on Wikipedia or bookmarked online) [8,9].

Academic visibility in the media has recently gained importance for reasons concerning both supply and demand. On the supply side, the Internet now provides academics with diversified, mobile, and easy-to-use web publishing tools and channels to disseminate their findings, including personal blogs and pages on platforms such as ResearchGate, LinkedIn, X, Academia.edu, Wikipedia and YouTube [4,6,9,10]. On the demand side, in today's new era when science must compete for public attention with fake news, "alternative facts", and pseudoscience [11,12], academics are more often asked or encouraged to present and discuss their findings in the media, to inform public debate and shed light on major issues (e.g., environment and climate, health and medical sciences, education, politics, technology).

Despite the general rise in academic visibility, research dissemination in newspapers, radio and television, news websites, and dedicated online platforms seems to make little room for women researchers. Across disciplines, men tend to be more active than women in communicating on social media [9] and personal blogs [13]. Women have made up some ground recently, but only 24% of expert voices heard in the news in 2020 were women, up from 19% in 2015 (Global Media Monitoring Project, FMPP 2020−2021 final report, https://whomakesthenews.org/wp-content/uploads/2021/08/GMMP-2020.Highlights_FINAL.pdf.). Women researchers' underrepresentation in the media is considered an enduring trend [4] with significant negative consequences. The extant research shows that this underrepresentation is detrimental to women academics' careers [14,15] but has not explored its impact beyond peers from the same discipline and academic institutions. Yet scientific dissemination efforts are increasingly directed toward broader lay audiences [6,9], including the general public, to foster scientific literacy and engagement, the educational community, to integrate research findings into teaching and training and decision-makers and socio-economic stakeholders, to inform policies and professional practices.

In 2021, Emmanuelle Charpentier expressed the wish that her Nobel Prize in Chemistry would *"provide a positive message to the young girls who would like to follow the path of science, and [...] show them that women in science can also have an impact through their research"* (https://twitter.com/NobelPrize/status/1624106077586350082). Her words suggest a belief that her own visibility could enhance both the perceived legitimacy of women researchers and women's confidence in their ability to pursue a successful academic career. The implies that promoting equal media visibility for women and men researchers could contribute to narrowing the gender gap in science. As Bastide [16] observes, women must first be seen and heard in order to gain respect and legitimacy, underscoring the importance of increasing their presence in the media.

Building on these perspectives, we posit that women researchers' lower visibility in scientific dissemination activities has the potential to reshape social representations of both research and researchers. In the present work, we investigate this phenomenon in the social sciences, focusing specifically on management science. We do so for two reasons. First, management science increasingly engages in research dissemination aimed at broad audiences and plays a central role in shaping organizational practices, leadership norms, and public discourse, making visibility dynamics in this field particularly consequential. Second, studying these dynamics in management science provides a conservative test: gender-science stereotypes are typically presumed to be weaker in the social sciences than in STEM and life sciences [e.g., 17, 18]. Demonstrating comparable visibility effects in this context therefore offers especially compelling evidence of the pervasiveness of gendered evaluative processes. Accordingly, we argue that these effects are likely to extend across all academic domains, from STEM to the social sciences.

In this article, we explore whether increasing the visibility of women researchers in scientific dissemination media within management science enhances audiences' perceptions of women researchers' expertise, self-efficacy beliefs, and the attractiveness of an academic career. To address this question, we conducted two laboratory experiments: the first with a sample of non-academic individuals (N = 271), representing the primary target of scientific dissemination activities, and the second with a sample of business school graduate students (N = 129), who are closer to the academic sphere and for whom role models may be particularly influential in shaping academic career aspirations.

## 2. Women researchers' lower visibility

To lay the foundations for our research, we synthesize prior work on the forms, causes, and consequences of women researchers' lower visibility, before deriving hypotheses regarding its differential effects on non-academic audiences and graduate students.

### 2.1. Forms of women researchers' lower visibility

Although the last few years have seen a rapid increase in the proportion of women academics, women remain relatively invisible in Science, Technology, Engineering, and Mathematics (STEM), especially in top academic positions [19,20]. In the US, 47% of doctoral degrees in 2023 were awarded to women (https://ncses.nsf.gov/pubs/nsf25300), but only 32.5% of professors were women in 2020 according to the American Association of University Professors. Their representation among full-time tenure-line faculty members decreases as rank increases, and cohort effects alone are not sufficient to explain their disappearance from academia as time passes [21], a state of affairs that has been described as *"the pipeline [...] leaking women"* [22].

Observable in most fields and countries [23], women researchers' lower visibility can take many forms [24]. Table 1 summarizes them according to the two levels of academic visibility [53]: the conceptual level and the literal level. At the conceptual level, the work of women researchers is less visible than their men counterparts' work. Women researchers appear to contribute less to the research process, publish less, and be cited less frequently. At the literal level, women researchers are physically less visible than their men counterparts. They speak less at scientific meetings and in the media, are less likely to be principal investigators (PI), raise less funding, engage less in innovative networks and patenting, receive fewer awards and, above all, find it harder to get jobs in academia.

Importantly, a substantial number of the studies reviewed in Table 1 (i.e., references marked with an asterisk), draw on data from the social sciences. This suggests that lower visibility is not confined to STEM or life sciences, where gender bias is often assumed to be most pronounced, but is also documented in disciplines typically perceived as more gender-balanced. The persistence of these patterns in the social sciences reinforces the argument that gendered visibility dynamics reflect broader evaluative processes operating across academic domains.

The different forms of academic visibility are connected [6]. Contribution visibility, reflecting publications and citations, is used to measure researchers' impact and justify hiring, retention, and promotion decisions [20,61,65,67,68]. It also

**Table 1. Forms of women researchers' lower visibility (references marked with an asterisk report data from the social sciences).**

| Level and form of visibility | | Research findings | References |
|---|---|---|---|
| **Conceptual level** | Publication and communication | Women publish less (especially highly competitive articles, i.e., articles with several co-authors and articles published in the highest-impact journals). They are also less frequently selected to give conference presentations. | [25*, 26,27*, 28*, 29*, 30*, 31,32*, 33*, 34*, 35*, 36*] |
| | Citation | Women are cited less frequently, partly because they cite themselves less often than men do. This is especially true for women with young children, and publications that focus on topics stereotypically associated with men. | [27*, 28*, 37*, 38–40] |
| | Contribution to the research process | Women are underrepresented in the high-status first and last author positions, and in single-authored papers. In medicine, they are less likely to be credited for the more prestigious parts of the research process (e.g., design, analysis, writing). | [28*, 29*, 41*, 42–44] |
| **Literal level** | Leadership | Women are less likely to be PIs and raise less funding. They hold fewer strategic positions in innovation collaboration networks. | [45–47*, 48] |
| | Collaborations | Women engage less with industry. Their collaborations are more domestically oriented. As graduate students, they are less likely to publish with senior colleagues of the opposite gender who produce high-impact research. | [28*, 49,50] |
| | Patents | Women are less involved in patenting. | [25*, 51,52] |
| | Questions | Women ask proportionally fewer questions when attending departmental academic seminars, lectures, presentations, or academic conferences, especially when the session is chaired by men and the first question is asked by a man from the audience. They are also underrepresented in asking the first question. | [53*, 54,55, 56,57] |
| | Invitations | Women are underrepresented among guest speakers at academic events and the experts consulted by the news media. | [4*, 9*, 13*, 58, 59*, 60*] |
| | Awards | Women receive fewer prestigious scientific awards. | [61,62*, 63,64] |
| | Positions | Women face greater challenges in getting chair positions, laboratory manager positions, and positions in professional societies and journal editorial boards. | [65*, 66] |

affects how often they are invited to speak at conferences and given awards and distinctions, two things that in turn make researchers more physically visible. More visible researchers benefit from more attention paid to their research, with obvious positive consequences in terms of citations and publishing opportunities that make their contributions more visible [15]. Academic visibility operates as a virtuous (or vicious) circle.

Focusing now specifically on researchers' visibility in the media, the next sub-sections discuss the causes and consequences of women researchers' lower media visibility.

## 2.2. Causes of women researchers' lower media visibility

The mechanisms at work in women researchers' lower media visibility are complex. They result from a combination of different causes that include not being asked to share their knowledge in the media, but also in many cases declining to do so.

A first apparent reason for women researchers' lower media visibility is believed to be a *"pervasive culture of negative bias − whether conscious or unconscious − against women in academia, resulting in a lack of professional support and networking"* ([69], p.986). This discrimination involves a range of psychosocial and structural mechanisms.

At the psychosocial level, discrimination is caused by implicit gender biases [70], i.e., stereotypical ideas about differences between women's and men's "natural" attributes or roles, which unconsciously affect perceptions and decisions [20,71]. This is in line with social role theory and its assumptions regarding gender differences [17]. Because, in traditionally gendered social roles, women appear more concerned with others' well-being and men appear more self-confident, competitive, and ambitious, there is an assumption that what is expected of individuals in a researcher's role is less compatible with female roles, and as a result science is more strongly associated with men than women [15]. Research of similar quality may even be assigned lower evaluations when credited to a women researcher [43,60,72], particularly if the topic is stereotypically associated with men and the evaluators have traditional attitudes to gender roles [67]. So even though the gender balance in science has improved in the last few decades [29,73], women researchers are likely to be less visible in the media because of implicit gender biases affecting perceptions of their research performance.

At the structural level, academic institutions, like any other type of organization [74], are not gender-neutral [75]. Their structures and male-dominated organizational culture have historically *"provided restricted employment, career, and leadership opportunities for women"* ([76], p.3]) and continue to marginalize women, impairing their academic visibility and careers. For instance, academic institutions do not *"take into consideration parenthood, family, and personal spheres of life"* [77]. They provide less institutional support for women researchers to access the resources typically needed in their discipline [26]. They are also characterized by an informal but powerful "Old Boys' Club effect" [34,78], not to mention the "Matilda effect" [79], named after Matilda Joslyn Gage, a 19th-century feminist and suffragist who pioneered the study of how women's contributions were often overlooked or attributed to men, which refers to the way men take credit for women's intellectual contributions. For these reasons, women researchers are expected to be less visible in the media because academic structures favor men researchers.

A second explanation for the relative media invisibility of women researchers may be their own self-silencing and inhibition [80], with women deliberately choosing to keep a low profile [81]. For example, women cite themselves less often than men do [82]. Also, when asked what prevents them from asking a question at academic seminars or conferences, they answer more frequently than men that they do not have enough time or nerve, feel they are not expert, feel intimidated by the speaker, or worry that they will not ask a good question or have misunderstood the content [21]. Similarly, many factors could account for their decision not to build a media presence, including higher levels of impostor syndrome and risk aversion when stepping out of their comfort zone, both in terms of their academic subject area and their media communication skills.

In line with social role theory [17], the stereotypical idea that brilliant researchers tend to be men [3] may erode women researchers' self-confidence. The relative absence of relevant media communication models for women researchers does not help. More practically, women researchers may also have less free time to devote to media work, as they often carry the main burden of caring for others. Not only do they manage household and family duties in addition to childbearing, women researchers also take more care of their academic "family" [83]: they perform significantly more services for their university, campus, or department (e.g., faculty governance and recruitment, evaluation and promotion, student admissions and scholarships, program supervision, development and marketing) than men researchers.

## 2.3. Impacts of women researchers' lower media visibility

The lower media visibility of women researchers has significant negative consequences for both the researchers themselves and society at large.

### 2.3.1. Personal impacts of women researchers' lower media visibility.
Among researchers' responsibilities [84,85], science outreach activities in the media have become increasingly important in evaluating research impact and scientific careers [8,9,86]. The first result of this is that women researchers' lower visibility in the media contributes to the gender gap in scientific careers. Second, and for the same reasons, that lower visibility also plays a significant role in academics' gender pay gap [2]. Third, gender stereotypes affect the way people treat others, but also how they perceive and define

 

themselves, their career ambitions, and their behavior. Negative stereotypes of women in science can harm women researchers' "*resolve to engage with domains (they) find personally valuable, undermine (their) ability to perform well, and impair (their) life outcomes.*" ([87], p.291). Lower media visibility for women researchers can therefore weaken not only their careers and salaries but also their overall self-esteem and well-being.

**2.3.2. Societal impacts of women researchers' lower media visibility.** Women researchers' invisibility affects the coverage of certain topics and dissemination of original findings, and that has performative effects on society at large. As recently shown, men and women favor different topics [88]. Women researchers tend to study more interdisciplinary, gendered and applied subjects [2,30], and use different ways of theorizing [89]. In management, women tend to focus more on social and human-centered areas, while men tend to gravitate towards technical and operational aspects [90]. Involving more women researchers in the media thus challenges prevailing conceptions of what constitutes appropriate and desirable management practices, and promotes a more diverse range of solutions that are more responsive to the challenges facing contemporary society. Also, in medicine, women researchers devote more attention to gender and sex analysis, which is increasingly being recognized as a key factor for enhancing medical research and healthcare [91]. In the end, increasing the visibility of women researchers' work leads to dissemination of better science, to the benefit of society [92]. However, the extant research concerning the impact on society of women researchers' lower media visibility does not document its impact on a non-academic audience nor on students' social representations of research and researchers.

**2.3.3. Conceptual framework.** In 1983, Chambers invited 4,807 children to participate in a "draw-a-scientist test" [93]. Only 28 women researchers were drawn, all by girls. Gender-science stereotypes caused by repeatedly observing that there are fewer women than men working in science appear at an early age, leading non-academics to form "*associations that connect science with men more than women*" [94, p.631]. As Eagly and her colleagues [95] note, gender-science stereotypes can lead people to believe that men make more competent researchers, being more intrinsically motivated to engage in research work as it is consistent with their culturally defined gender roles. Women working in research, on the other hand, may be considered to be violating culturally-defined gender roles and attract prejudiced reactions, including biased evaluations of their competence and negative preconceptions about their performance.

However, these stereotypes can be changed through repeated exposure to counter-stereotypic portrayals of women in research [94]. Increasing women researchers' visibility in the media could thus weaken gender-science stereotypes, loosen the association of science with men rather than women, and finally reduce the perception that men are more competent at research.

We therefore formulate Hypothesis 1 as follows:

**H1**: Greater women researchers' visibility in scientific dissemination media will lead to higher perceived expertise of women researchers compared to lower women's visibility.

Going further, strong gender-science stereotypes increase identification with science and science career aspirations in men, but decrease them in women, due to "stereotype threat" [96]. Yet when exposed to women researchers, women identify more with science and stereotype science as more feminine than masculine, and this increases their science identification and career aspirations, which could impact their professional trajectories. In chemistry, women students working with women mentors tend to be more productive during their PhD and are more likely than women students with men mentors to become faculty members themselves [97]. Women students or junior scholars need to have female role models, i.e., "*individuals whose behaviors, personal styles, and specific attributes are emulated by others*" ([98], p.52) for reassurance that it is possible for women to reach certain positions [99].

Social learning theory [100] states that experiences gained vicariously through peer modeling are an important way to establish and reinforce self-efficacy beliefs, whether regarding mathematics, science, leadership, sexuality, or career choice (e.g., [101–104]). Specifically, subjects' identification with and positive attitude toward a role model are likely to raise self-efficacy beliefs [105,106]. However, such identification depends on perceived similarity with the role model. This

suggests that exposure to successful female role models will only positively influence self-efficacy beliefs for imitative attitudes and behaviors in a same-gender audience [106]. Therefore, making female role models more visible in academia, including in the media, could provide successful ingroup role models, boost self-efficacy, signal to young women that they could also have a successful academic career, and avoid limiting their ambitions [73,107], with potentially the opposite impact on young men.

We therefore formulate Hypothesis 2 as follows:

**H2**: Gender moderates the influence of women researchers' visibility in scientific dissemination media on self-efficacy beliefs, such that greater visibility increases self-efficacy beliefs among women but decreases them among men.

Regarding the influence of women researchers' media visibility on the perceived attractiveness of an academic career, we argue that the devaluation theory may operate alongside the role modeling theory. According to the devaluation theory [108–110], the perceived value of an occupation is gendered: society tends to attribute higher status and prestige to male-dominated professions compared to those dominated by women [111]. This perspective is particularly relevant in educationally advantaged environments [112], where occupational prestige is often closely monitored. Consequently, increasing the media visibility of women researchers, although intended to inspire, may paradoxically diminish the perceived value of an academic career among both young men and women. For young women, this creates a tension between two opposing mechanisms: while the role modeling theory predicts a positive effect of greater women researchers' visibility on career aspirations, the devaluation theory implies a potential negative backlash. For young men, both theories converge in predicting a negative effect, as greater women researchers' visibility may reduce the perceived status of the academic profession and provide fewer identity-relevant role models at the same time. Ultimately, regardless of whether the net effect on young women is positive or neutral, this dual-theory framework highlights the importance of gender as a moderating factor in understanding how women researchers' visibility shapes the perceived attractiveness of academic careers. We therefore formulate Hypothesis 3 as follows:

**H3**: Gender moderates the influence of women researchers' visibility in scientific dissemination media on the perception of academic career attractiveness, such that greater visibility increases the perceived attractiveness of an academic career among women but decreases it among men.

Fig 1 displays our conceptual framework.

To test our conceptual framework, we used two between-subjects laboratory experiments. Although such experiments are quite rare in previous research on biases resulting from scholars' gender [67], for a thorough examination of gender biases [28], it is essential to be able to control the materials being evaluated when randomly assigning participants to conditions that differ only in the gender presented.

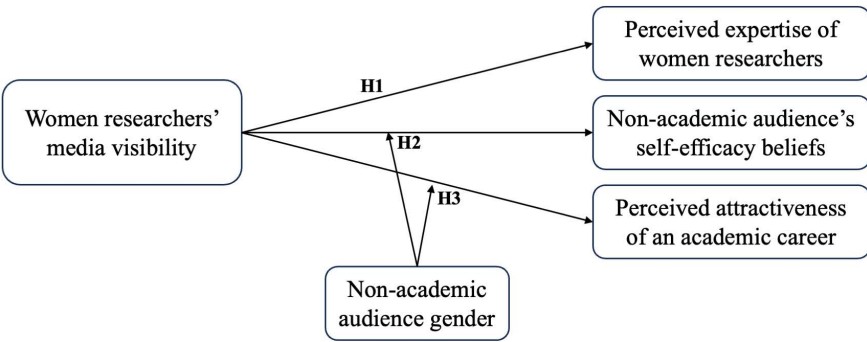

**Fig 1. Conceptual framework.**

## 3. Study 1

Study 1 uses a sample of non-academic participants to test H1 and H2.

### 3.1. Method

This experiment was approved by the Ethics Committee of one of the researchers' institutions: project titled "*Effets de la représentativité des femmes dans la diffusion des connaissances en management*" was approved by the Comité d'Ethique de la Recherche of HEC Montréal on November 23, 2021, under the approval number 2022–4756 (Nagano form F1-16341). Participants in this study were recruited on a voluntary basis and completed an online experimental questionnaire. Data was collected between Feb 4th 2022 and Nov 2nd 2022. As no personally identifying information (such as IP addresses or names) was collected, the data are fully anonymous. In accordance with the approved ethics protocol, informed consent was obtained through the completion of the online questionnaire, which included clear instructions and served as the basis for implicit written consent.

We recruited 347 French participants through the online panel of PanelLab, a European professional market research institute. Recruitment materials described the study as a short research survey on media and decision making, without mentioning gender or academia to avoid priming participants. Participants gave informed written consent for the experiment.

They were then exposed to a gender-biased newsletter about researchers' contributions to the field of management, which randomly presented a large majority of men vs. women researchers (see Fig 2).

The design of our experimental stimuli was inspired by actual newsletters distributed by online platforms that aim to popularize academic research (e.g., Xerfi Canal on YouTube, which displays more than 116k followers as of August 12, 2025). To maintain ecological validity, the men-dominated condition, featuring eight men researchers out of nine, mirrors real patterns of research dissemination observed in one of the authors' home countries. This condition serves as our control. To eliminate potential confounds related to visual identity, all researcher photos were generated using AI (www.thispersondoesnotexist.org, a site that grants permission to use the generated faces for personal, commercial purposes.). We selected management as the academic field featured in the newsletter because it occupies a middle ground in terms of gender participation, differing markedly from STEM fields such as engineering, physics, or computer science, where men are the clear majority. Moreover, we chose research topics that are neutral in terms of gender associations, in line with findings by Eom and colleagues [113], who highlighted the potential for interaction effects between gender cues and the perceived gender-typicality of research domains. Across conditions, the only manipulated variable was the gender of the researchers.

To simulate deliberate rather than incidental media exposure, we explicitly instructed participants to read the newsletter content carefully, as they would be asked questions about its content afterward. We acknowledge that this structured exposure differs from organic media consumption; however, it is common and necessary in experimental settings where stimulus control is essential. After exposing participants to our stimuli, we measured their general self-efficacy beliefs. We then showed them a presentation of a research project being undertaken by "Clémence Dubois", a fictitious woman researcher. It contained a short description of her fictitious project and a picture generated by the same AI used for the initial stimuli (see Fig 3). We then measured participants' perceptions of Clémence Dubois' expertise.

As displayed in Table 2, all items were drawn from established literature. They were measured on 7-point scales ranging from "*Not at all*" to "*Extremely*" and displayed satisfactory psychometric properties.

Towards the end of the survey, participants were asked to indicate their gender ("man," "woman," "non-binary," "prefer to self-describe [free-text field]," or "prefer not to say"), age, and highest educational qualification. They were also asked about the gender balance ("rather men-dominated," "rather women-dominated" or "I do not remember") of the researchers featured in the newsletter at the beginning of the survey, which served as the post-hoc manipulation check for our experiment. At the conclusion of the survey, participants were debriefed and informed that all the research presented was fictitious and that the researchers shown did not exist.

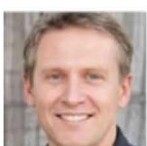

Rethinking M&A strategies : Symbiosis integrations

**Dominique Roy**

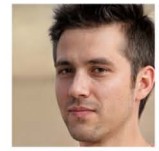

God in the face of research and teaching in management

**Camille Dupont-Lieu**

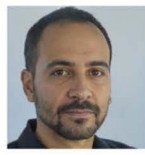

The 4 figures of the manager : the magician, the tyrant, the master, the parent

**Louis Michel**

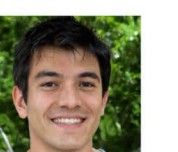

Public innovation and new forms of public management

**Claude Rival**

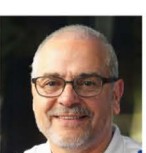

The crises are accelerating but do not modify consumption trends

**Alexandre Sion**

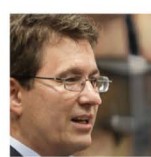

Competitive dynamics : Singular actors, plural strategies

**René Strauss**

Higher management education and French-influenced diplomacy : what future ?

**Frédéric Lepetit**

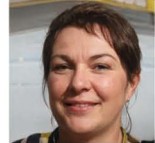

For research in management, connected to today's challenges

**Augustine Bergeron**

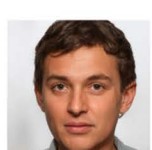

Changes in logistics

**Michel Tiers**

---

The management research newsletter

September 1, 2021

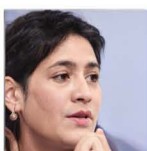

Rethinking M&A strategies : Symbiosis integrations

**Dominique Roy**

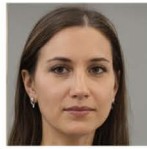

God in the face of research and teaching in management

**Camille Dupont-Lieu**

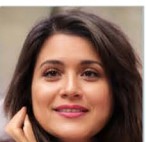

The 4 figures of the manager : the magician, the tyrant, the master, the parent

**Louise Michel**

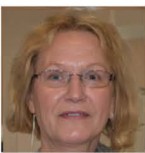

Public innovation and new forms of public management

**Claude Rival**

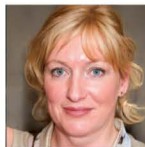

The crises are accelerating but do not modify consumption trends

**Alexandra Sion**

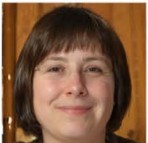

Competitive dynamics : Singular actors, plural strategies

**Renée Strauss**

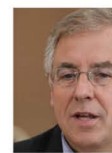

Higher management education and French-influenced diplomacy : what future ?

**Frédérique Lepetit**

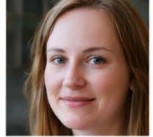

For research in management, connected to today's challenges

**Augustine Bergeron**

Changes in logistics

**Michelle Tiers**

**Fig 2. Stimuli (on the left: the men-dominated research newsletter/ on the right: the women-dominated research newsletter).**

**Leadership in the creative industries in Quebec**

This research project explores the key skills that leaders and managers of Quebec's creative industries must develop to deal with the specificities of these very varied industries (video games and perfume, haute cuisine and music, publishing and fashion, performing arts and architecture, etc.). In particular, Professor Clémence Dubois explores the ways in which creative professionals learn and highlights the role that context and culture play in managing the tensions between creativity and economic return within these industries.

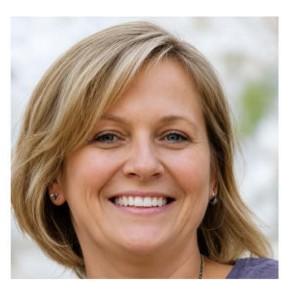

**Research led by Clémence Dubois, PhD, Professor of Management**

**Fig 3. Presentation of a fictitious woman researcher and project.**

**Table 2. Psychometric indicators for scales (Study 1 and Study 2).**

| Factors | Items | Factor loadings | | AVE | | Composite reliability | |
|---|---|---|---|---|---|---|---|
| | | S1 | S2 | S1 | S2 | S1 | S2 |
| S1: General self-efficacy beliefs [114]. | If I am in trouble, … | | – | .759 | – | .927 | – |
| | I can usually think of a solution | .897 | | | | | |
| | I can always manage to solve difficult problems if I try hard enough | .863 | | | | | |
| | I can usually handle whatever comes my way | .869 | | | | | |
| | I know how to handle unforeseen situations | .856 | | | | | |
| S1 & S2: Perceived expertise [115] | To what extent would you say that Clémence Dubois is… | | | | | | |
| | an expert in leadership | .903 | .906 | .815 | .839 | .946 | .955 |
| | experienced in leadership | .889 | .894 | | | | |
| | knowledgeable about leadership | .910 | .942 | | | | |
| | qualified to talk about leadership | .909 | .920 | | | | |
| S2: Perceived attractiveness of an academic career (*ad hoc*). | To what extent… | – | | – | .796 | – | .922 |
| | would you seriously consider an academic career? | | .894 | | | | |
| | would the research environment be a good working environment for you? | | .882 | | | | |
| | is an academic career attractive to you? | | .900 | | | | |
| S2: Specific self-efficacy beliefs [116]. | To what extent do you feel confident in your ability… | – | | – | .659 | – | .853 |
| | to express your opinion on a management issue | | .800 | | | | |
| | to discuss economic policy with your friends | | .777 | | | | |
| | to defend your position on contemporary organizational issues | | .856 | | | | |

We excluded from the analysis 3 participants who did not identify as either a man or a woman in order to focus on gender binary categories and 73 participants who provided an incorrect response regarding the gender composition of the newsletter. Interestingly, more participants were wrong about having seen a men- (vs. a women-) dominated newsletter (51 out of 169 vs. 22 out of 175; $\chi^2_{(1)}$=15.94, p < .001). This indicates that overrepresentation of men in research, including in the field of management, is less unusual and less noticeable than overrepresentation of women in research, which is consistent with the underrepresentation of women to date in all leading management and organization studies journals [30]. The final sample thus comprised 271 participants (mean age 42.4, 50.6% men) of whom 24% had non-university education and 76% held university degrees. Participants in the two conditions did not differ in terms of gender ($\chi^2_{(1)}$=.28, p = .59), age ($F_{(1,269)}$=.01, p = .97), or educational qualifications ($\chi^2_{(3)}$=1.74, p = .63).

## 3.2. Results

We first tested the impact of our manipulation on participants' general self-efficacy beliefs. An ANCOVA controlling for their gender, age, and qualifications showed no main effect caused by the manipulation ($F_{(1,266)}=.07$, $p=.79$), but found a main effect of gender ($F_{(1,265)}=6.30$, $p=.01$, $\eta_p^2=.02$) along with an interaction effect with gender ($F_{(1,265)}=3.87$, $p=.05$, $\eta_p^2=.01$). The main effect of gender is in line with a recent meta-analysis [117], that found gender differences in self-esteem, which was higher in men, especially in more developed countries. Regarding the interaction effect, women participants reported higher self-efficacy beliefs when exposed to a majority of women vs. men researchers, while the opposite was observed for men, corroborating H2 (see Fig 4a). Interestingly, in our experiment the gender difference in self-efficacy beliefs observed in response to a men-dominated panel ($F_{(1,115)}=9.08$, $p<.01$, $\eta_p^2=.07$) disappears when participants are exposed to a women-dominated panel ($F_{(1,148)}=.14$, $p=.71$).

We then tested the impact of our manipulation on participants' perceptions of Clémence Dubois' expertise. An ANCOVA, controlling for the same variables as previously, showed that our manipulation had an effect ($F_{(1,266)}=9.76$, $p<.01$, $\eta_p^2=.03$). Specifically, Clémence Dubois' perceived expertise was higher when participants were exposed to a women-dominated panel ($M=4.79$) than a men-dominated panel ($M=4.39$) (see Fig 4b). No interaction with gender was found ($F_{(1,265)}=.05$, $p=.82$). These results support H1.

Results from Study 1 suggest that increasing the visibility of women researchers in scientific dissemination media can enhance perceptions of their expertise and reduce gender gaps in self-efficacy beliefs, highlighting the potential of such media to shape public perceptions. Study 2 aims to assess the robustness of these effects by examining whether individuals more familiar with the academic world of management, namely business school graduate students, are less susceptible to biases stemming from women researchers' invisibility. It also allows us to test H3, which could not be examined in Study 1, as participants drawn from the general non-academic population cannot meaningfully report intentions to pursue an academic career.

# 4. Study 2

## 4.1. Method

This experiment was approved by the Ethics Committee of one of the researchers' institutions: project titled "*Effets de la représentativité des femmes dans la diffusion des connaissances en management*" was approved by the Comité d'Ethique de la Recherche of HEC Montréal on November 23, 2021, under the approval number 2022–4756 (Nagano form F1-16341). Participants in this study were recruited on a voluntary basis and completed an online experimental

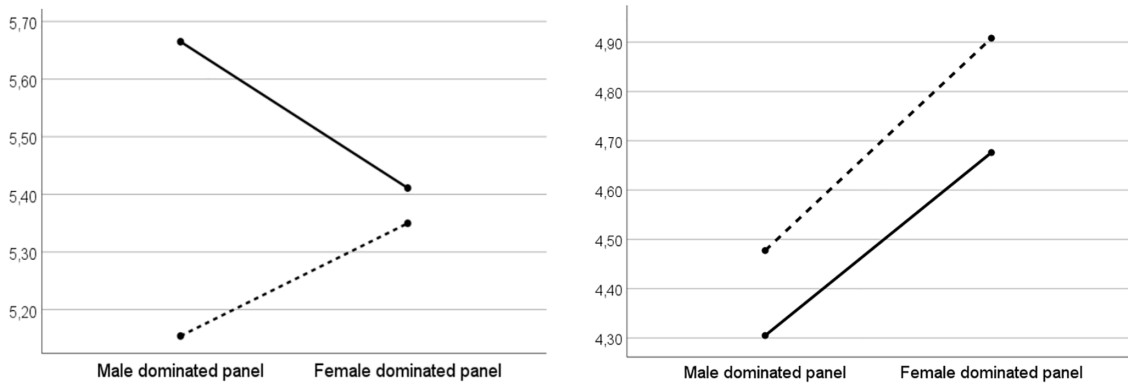

**Fig 4. a.** Impact on self-efficacy beliefs (women: dotted line/ men: solid line). **b.** Impact on perceived expertise (women: dotted line/ men: solid line).

questionnaire. Data was collected between Feb 4th 2022 and Nov 2nd 2022. As no personally identifying information (such as IP addresses or names) was collected, the data are fully anonymous. In accordance with the approved ethics protocol, informed consent was obtained through the completion of the online questionnaire, which included clear instructions and served as the basis for implicit written consent.

We recruited 148 graduate students enrolled in a business school (HEC Montréal) to participate in Study 2. Participants were contacted through the school's graduate student mailing list, completed the survey voluntarily without any incentive, and responded in French. Study 2 followed the same protocol as Study 1, with two notable exceptions. First, after exposing participants to our stimuli, we measured the perceived attractiveness of an academic career using 3 *ad hoc* items (e.g., "*To what extent do you feel academia would be a good work environment for you?*"). Second, we adapted our self-efficacy beliefs measure for an academic context, to check whether the result of Study 1 was robust to the measure used to assess the construct. Clémence Dubois' perceived expertise was measured using the same items as in Study 1. As in Study 1, the scales displayed satisfactory psychometric properties (see Table 2).

We excluded from the analyses 3 participants who did not identify as either a man or a woman and 16 participants who provided an incorrect response regarding the gender composition of the newsletter. In Study 2, only one participant did not check the manipulation (i.e., did not correctly recall whether the newsletter stimulus predominantly featured men or women), compared with 73 participants in Study 1, a noteworthy difference in itself. The final sample comprised 129 participants (mean age 28.8, 29.5% men). Participants in the two conditions did not differ in terms of gender ($\chi^2_{(1)}$=1.21, p=.27) or age ($F_{(1,127)}$=.02, p=.88).

## 4.2. Results

We first tested the impact of our manipulation on participants' self-efficacy beliefs with respect to speaking about management and economic issues. An ANCOVA controlling for gender and age showed no main effect caused by the manipulation ($F_{(1,125)}$=.52, p=.47), but found an interaction effect with gender ($F_{(1,124)}$=6.08, p=.01, $\eta_p^2$=.05). Replicating H2, women graduate students report higher self-efficacy beliefs when exposed to a women (vs. men)-dominated panel of researchers, while the opposite is observed for men graduate students (see Fig 5a). The expected gender difference observed in the self-efficacy beliefs of participants exposed to a men-dominated panel ($F_{(1,62)}$=6.86, p=.01, $\eta_p^2$=.10) disappears in the case of exposure to a women-dominated panel ($F_{(1,61)}$=1.59, p=.21).

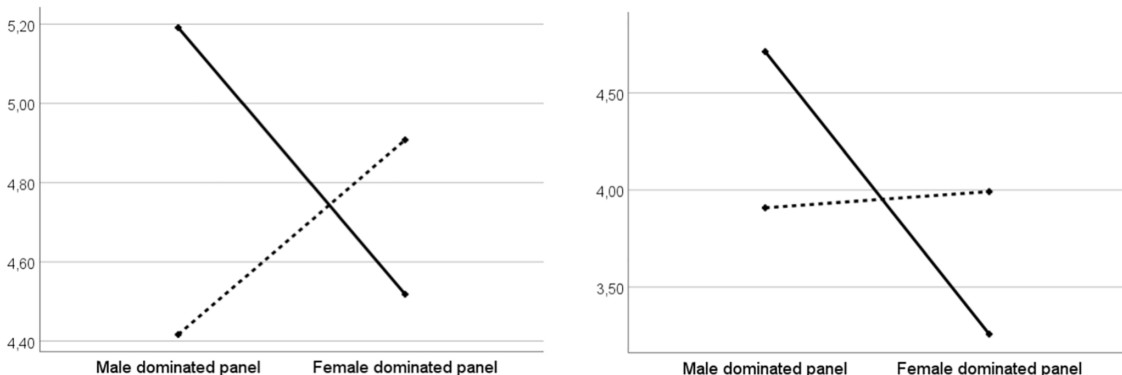

**Fig 5. a. Impact on self-efficacy beliefs (women: dotted line/ men: solid line), b. Impact on perceived attractiveness of the academic career (women: dotted line/ men: solid line).**

We then tested the impact of our manipulation on participants' perceptions of Clémence Dubois' expertise, and found no main effect ($F_{(1,125)}$=.35, p=.56), nor any interaction effect with gender ($F_{(1,124)}$=.47, p=.49). In this regard, Study 2 does not replicate H1.

Finally, we tested the impact of our manipulation on participants' perceptions of the attractiveness of an academic career, and found no main effect ($F_{(1,125)}$=1.31, p=.25), but an interaction effect with the graduate students' gender ($F_{(1,124)}$=5.22, p=.02, $\eta_p^2$=.04). As shown in Fig 5b, men perceive the academic profession as less attractive when they are exposed to a majority of women vs. men researchers ($F_{(1,35)}$=7.88, p<.01, $\eta_p^2$=.18). However, our data do not demonstrate the reverse trend among students of the opposite gender ($F_{(1,88)}$=.06, p=.81). Therefore, H3 is fully supported.

## 5. Discussion

This research examines a form of academic visibility that is becoming increasingly common and valued in academic circles, namely researchers' visibility in scientific dissemination media within the social sciences. Drawing on social role theory of sex differences [17,18], we adopt an experimental approach to establish clear causal links. Our results provide initial evidence that exposure to curated academic content in a newsletter format can shape perceptions among both non-academic audiences and graduate students, yielding meaningful theoretical insights and several actionable recommendations.

### 5.1. Theoretical contributions

Our research makes two key contributions to the literature on the gender gap in science.

First, it extends the scope of the impacts of management women researchers' lower visibility. While cross-disciplinary studies have mainly emphasized the personal career costs associated with women's underrepresentation in the media, such as fewer professional opportunities [8,9], our findings reveal additional consequences that extend beyond academia. Its results show that the lower visibility of women researchers in the media can fuel a large gender gap in self-efficacy beliefs in the non-academic population and in the graduate student population, suggesting that even a specific form of research dissemination, such as a newsletter highlighting academic experts, can shape public perceptions in meaningful ways. While our study does not encompass the full range of media, it points to the potential societal impact of gendered representation in social sciences academic communication. This finding is of major importance, as self-efficacy beliefs naturally result in self-limiting behaviors [118]. Going further, limited representation of management science women scholars in structured research communication may contribute to the persistence of gender-science stereotypes implying that women researchers are less competent. Proximity to the academic world is identified as a relevant antecedent of the salience of gender-science stereotypes: making women researchers more visible may enhance perceptions of their expertise among the general population, but not among graduate students. With their greater knowledge of academia and how it works, graduate students are probably less susceptible to gender-science stereotypes.

Second, our research contributes to the literature on the gender gap in science by showing that the gender gap in academics' media visibility is central to understanding the gender gap in scientific careers. Specifically, management scholar women researchers' lower media visibility can weaken young women's self-efficacy beliefs before they have even embarked on an academic career and may lead them to feel less legitimate in aspiring to such a career. Once they have embarked on an academic career, awareness that they may be perceived as less expert than their counterparts because of their lower visibility could undermine their confidence and potentially hinder their long-term advancement. These findings contribute to an understanding of the accumulative advantages at work in the world of research where *'to those who have, more will be given'* [72,119]. They are also a response to the European Commission's [23, p. 21] call for more research on "*the complex mix of structural barriers, discrimination and cumulative disadvantages that account for women's underrepresentation in the highest scientific positions*".

Our findings also bring new insights to the role modeling theory by showing that making women researchers more visible is associated with a diminution of self-efficacy beliefs among men and a reduction of perceived academic career attractiveness among students of the opposite gender. This result diverges from previous research that found no significant effect of seeing a woman scientist on the self-efficacy [107,120] or career aspirations [121] of students of another gender. However, these studies differ in key ways from our own research design. For instance, both Stout and colleagues' Study 3 [107] and Young and colleagues [121] investigated the impact of long-term exposure to men or women professors in classroom settings. Such sustained interactions offer students familiarity with the role model and likely engage different cognitive processes than those triggered by brief exposure to a media-based expert, which tends to elicit more stereotyped perceptions. Moreover, the way self-efficacy was measured in these studies is critical. In their third study, Stout and colleagues [107] assessed self-efficacy using students' expected grades in a course, i.e., an indicator that may be heavily influenced by anticipated grading behavior of men versus women instructors, and that does not directly capture perceived personal competence. In the work of Young and colleagues [121], self-efficacy was assessed via participants' belief in their ability to become like the target in the future, which may be confounded by gendered attributions such as perceived competence or likability of the role model. Taken together, these methodological and contextual differences may explain why prior studies found no effect of women expert exposure on students of another gender, whereas our findings reveal a measurable decline in self-efficacy and academic career attractiveness. By focusing on short-term media exposure and employing more targeted measures of self-efficacy, our research calls to explore the conditions in which men respond negatively to women role models, and in particular the influence of the familiarity with the role model.

Finally, our findings contribute to the understanding of an occupation's gender composition and judgements of its attractiveness by highlighting the importance of an integrated theoretical framework that combines role modeling theory and devaluation theory. The literature suggests that the presence of a single female role model can be particularly effective in enhancing women's career aspirations, consistent with the predictions of role modeling theory [122]. However, our findings show that this positive effect may be undermined when women candidates are exposed to a majority of women role models, which may inadvertently trigger devaluation mechanisms. In such cases, the perception that a profession has a predominantly women workforce may lead to its reduced social valuation, thereby offsetting the motivational benefits associated with exposure to role models. These results underscore the importance of how role models are presented and activated in shaping career-related attitudes and decisions. The effectiveness of role models appears to depend not only on their presence but also on the way they are framed. In the end, combining both theories in future research could illuminate the inconclusive findings of previous research on the relationship between an occupation's gender composition and judgements of its attractiveness [119].

### 5.2. Recommendations

The European Commission [23] writes that gender segregation in scientific career advancement *"is clearly at odds with the scientific ethos of universalism and meritocracy: if universalism and meritocracy were the actual rules, gender inequality would be less prevalent than in other professions."* The advancement of women in science remains slow, fragile, and structurally constrained, and should not be taken for granted. It is unacceptable that lower media visibility further amplifies the systemic obstacles faced by women researchers. Enhancing women's media presence is both a matter of fairness and a strategic lever to foster inclusive, plural, and credible representations of scientific expertise. A coordinated, multi-stakeholder effort is essential to achieve this objective.

To academic institutions, we recommend to proactively encourage and support women researchers' engagement with the media. This involves allocating time and resources for training, formally recognizing public communication in promotion criteria, and creating psychological safety for women engaging in public debate.

Regarding platforms dedicated to improving women's visibility in the public sphere and media, we think that they should be strengthened: they must develop and maintain searchable directories of women experts, offer tailored media coaching,

and facilitate connections with journalists. Examples include SheSource, a global database of women experts created by the Women's Media Center; SourceHer, a curated database of African women experts; and the Columbia Journalism Review's list, which includes women, nonbinary people, and people of color across sectors. Moreover, registering as an expert on these platforms should be quick and simple to encourage as many women researchers as possible to list themselves and their areas of expertise.

Our recommendations extend to researchers. We encourage women researchers themselves to seek support and inspiration through social media (e.g., women.doing.science on Instagram, 500womensci on Twitter) and join the aforementioned platforms. Women researchers may also benefit from the GIJN Guide, a comprehensive resource for women journalists that includes expert directories, mentoring platforms, and safety protocols. Researchers, regardless of gender, who decline media invitations should adopt the reflex of referring journalists to the most competent woman on the subject.

Finally, media and communication professionals, including those within universities, have a responsibility to reshape the norms of representation, because women researchers' low media visibility shapes and sustains a gender-imbalanced picture of society and affects our self-conceptions through the harmful gender-science stereotypes they convey or reinforce [123,124]. Furthermore, these stereotypes have a negative impact on women's leadership self-perceptions and aspirations [125]. As the French Council for Advertising Ethics has observed, *"any form of communication that keeps individuals in "assigned" roles for [...] a given gender […] must be questioned, since it is part of an environment of stereotypes marked by social relationships that are generally hierarchical and non-egalitarian"* (our own translation). The promotion of higher media visibility for women researchers should also stop using illustrations depicting women researchers as *"grumpy or sexy, over-emotional or cold, victimized or bitchy"* [126] and simply present them as expert professionals [127]. Media standard-setters should therefore 1) require gender parity in expert panels and interviewee selections; 2) make a real effort to identify who is currently the leading expert on the subject covered, rather than always reverting to and sharing the same old contact list of men researchers; 3) invest time in preparing women researchers for interviews, offering guidance and reassurance to enhance their perceived behavioral control and reduce hesitancy; and 4) systematically record, monitor, and publish gender-disaggregated statistics on expert participation in media outputs.

This last point underscores the urgent need for gendered metrics and visibility audits. Such practices echo initiatives like the BBC's 50:50 Project, which demonstrates that transparency and self-monitoring can drive structural change. As summarized by Lord Kelvin's often-misquoted dictum, *"when you can measure what you are speaking about, and express it in numbers, you know something about it"* [128]. To measure is to know.

Yet, we must acknowledge the complexity of these dynamics. Our research indicates that increasing the media visibility of women researchers, while reducing gender-science stereotypes for some, may also inadvertently reduce self-efficacy and academic career attractiveness among students of the opposite gender. This paradox highlights the need for a balanced and reflexive approach to science dissemination, ensuring that representation strategies do not foster new asymmetries. Ultimately, these findings are an impetus for academics to open space for critical conversations in the classroom, about media, gender, credibility, and the implicit norms shaping academia. If science is to remain credible, it must not only be inclusive but be seen as inclusive.

### 5.3. Limitations and future research

Beyond its theoretical contributions and implications, this research opens up many avenues for future research. First, our experiments could be replicated in other cultural settings, or using different gender mixes in the research dissemination newsletter, to identify a more socially beneficial distribution in terms of representations of gender, research, and researchers. A further limitation is that we examined these dynamics exclusively within the field of management. Although this focus limits generalizability, it also constitutes a substantive contribution, as gender bias is often presumed to be more salient in STEM and life sciences. Showing that similar dynamics operate in management and the social sciences—where such stereotypes are assumed to be weaker—offers particularly informative insights. The experimental design could

therefore be extended to other forms of media, such as podcasts, television programs, or popular science magazines, in order to test whether the observed effects hold across different dissemination channels. Likewise, applying a similar approach to other academic disciplines beyond management, such as natural sciences, engineering, medicine, or the arts, could reveal whether the dynamics we observe are discipline-specific or more generalizable.

The design could also be extended to explore the moderating influence of skin color or disability and thus address questions of an intersectional nature, as we know that women belonging to underrepresented minority groups have to cope with a complex blend of sexism, racism, and other factors such as ableism (as seen in bias against autistic scientists) in STEM [129,130]. A further limitation of the present study is that we did not collect data on participants' race/ethnicity or first-generation college status. These social positions may intersect with gender to shape perceptions of researchers' visibility, self-efficacy beliefs, and the attractiveness of academic careers. Future research would therefore benefit from adopting a more explicitly intersectional perspective to examine how multiple dimensions of identity jointly influence responses to scientific representation in the media. Finally, future studies could investigate institutional or communicative mechanisms that may encourage greater media engagement among women researchers.

At this stage, our study contributes by rendering visible women's invisibility in research dissemination activities and by highlighting some of its consequences for both non-academic audiences and graduate students. Who communicates research in scientific dissemination media within the social sciences matters, as visibility establishes implicit norms and beliefs about who succeeds in academia, with far-reaching implications for both society at large and academic communities themselves. If media shape our representations, and representations in turn shape society [20], then it is time to collectively reflect on the representations we promote, and to take concrete steps to change them.

## Supporting information

**S1 File. S1 for PLOS One (N=271).**
(SAV)

**S2 File. S2 for PLOS One (N=129).**
(SAV)

## Acknowledgments

The author would like to express sincere gratitude to Denis Grégoire for his insightful feedback and valuable suggestions, which greatly contributed to improving this work. The author also warmly thanks the entire M-Lab team at Paris Dauphine for their constructive comments and continuous encouragement throughout the development of this article.

## Author contributions

**Conceptualization:** Beatrice Parguel, Marine Agogué, Julie Gauneau.

**Data curation:** Beatrice Parguel, Marine Agogué.

**Formal analysis:** Beatrice Parguel, Marine Agogué.

**Investigation:** Beatrice Parguel, Marine Agogué.

**Methodology:** Beatrice Parguel, Marine Agogué.

**Project administration:** Beatrice Parguel, Marine Agogué.

**Resources:** Beatrice Parguel, Marine Agogué, Julie Gauneau.

**Software:** Beatrice Parguel.

**Supervision:** Beatrice Parguel, Marine Agogué.

**Validation:** Beatrice Parguel, Marine Agogué.

**Visualization:** Beatrice Parguel, Marine Agogué.

**Writing – original draft:** Beatrice Parguel, Marine Agogué, Julie Gauneau.

**Writing – review & editing:** Beatrice Parguel, Marine Agogué.

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
