## [Decision Letter · Decision Letter 0]

10 Jul 2025

PONE-D-25-25301When Matilda shows up: The double-edged impact of female researchers’ media visibility on the non-academic audiencePLOS ONE

Dear Dr. Agogué,

Thank you for submitting your manuscript to PLOS ONE. After careful consideration, we feel that it has merit but does not fully meet PLOS ONE’s publication criteria as it currently stands. Therefore, we invite you to submit a revised version of the manuscript that addresses the points raised during the review process.

We look forward to receiving your revised manuscript.

Kind regards,

Robin Haunschild

Academic Editor

PLOS ONE

3. We note that Figure 2 and 3 includes an image of a participant in the study.

Reviewers' comments:

Reviewer's Responses to Questions

**Comments to the Author**

1. Is the manuscript technically sound, and do the data support the conclusions?

Reviewer #1: Partly

Reviewer #2: Yes

2. Has the statistical analysis been performed appropriately and rigorously? 

Reviewer #1: Yes

Reviewer #2: Yes

3. Have the authors made all data underlying the findings in their manuscript fully available?

Reviewer #1: No

Reviewer #2: Yes

4. Is the manuscript presented in an intelligible fashion and written in standard English?

Reviewer #1: Yes

Reviewer #2: Yes

5. Review Comments to the Author

Reviewer #1: This paper aims to assess whether increasing the presence of women in the media affects self-efficacy and perceived expertise among academic and non-academic audiences, as well as assessing the impact on graduate student career interests. Overall, this paper is technically sound. However, many of the conclusions are overreaching and the language should be tempered throughout. It is important to note that this was done in one specific context (management science) and the results should not be generalized to science or STEM more broadly. Similarly, one type of media representation (a newsletter) was examined. As such, this also should not be generalized to “media” more broadly. Finally, the authors use terms for sex (female) and gender (woman) interchangeably. I would recommend focusing the language on the paper on women- the social construct of gender is more relevant here than the biological sex of being female. My specific comments are below.

Abstract

• The first sentence uses ‘women,’ which indicates gender and ‘female’ which indicates sex. Please investigate how these terms are different and consider which is more appropriate for this study. I think that using terms that reflect gender is most important. People who appear as women have an impact on others who also identify as women.

• It is unclear whether the gains (increased perceived expertise and decrease the gender gap for self-efficacy beliefs) is specific to the public, grads, or both.

• Please include null results in the abstract as well.

• Temper language to ensure that “management science” is not being generalized to “science” and that a “newsletter” is not be generalized to all “media.

Section 1

• In section 2 it seems odd that the impact on an academic audience (graduate students) is not mentioned here given the sample focused on that population.

• End of section 1: It seems a bit unusual to summarize the findings here.

Section 2.2

• The word “refusing” seems strong here and perhaps an overstatement. “Decline” may be more appropriate.

Section 2.3

• Section 2.3.1 The sentence that says that lower media visibility weakens careers, salaries, self-esteem and wellbeing is a bit too strong. I suggest rephrasing by adding “can” weaken.

• 2.3.3: It is unclear what is meant by counter-stereotypical women in research. This implies that the woman herself is counterstreotypic.

• Paragraph after H1 (pg 10) sentence in the middle of a paragraph abruptly cuts off.

Section 3.1

• What is management science? Is this considered STEM? As a reviewer in the US, I was unfamiliar with this field and think many readers would appreciate a brief explanation and also what distinguishes it as a STEM field.

• Typo near citation 121

• How were graduate students recruited specifically? What language was used in the recruitment for the study? How were participants incentivized?

• In what ways were the participants told to interact with this newsletter? Presumably no lay person is going to randomly interact with a newsletter about management science. As such, how do you view this as an exercise that reflects lay people’s interaction with the media more broadly?

• The paper should acknowledge the presence of gender nonbinary and gender queer individuals. The paper can acknowledge their existence in science while still focusing on gender binary categories. Relatedly, were participants given the option to identify as gender non-binary?

Section 4

• The introduction should be framed with this experiment in mind. Currently, these experiments are largely absent from the intro.

• I strongly disagree that “To what extent do you feel confident in your ability to express your opinion on a management issue” is reflective of perceived attractiveness of an academic career. Is this a previously validated scale to measure attractiveness of an academic career? I think this is a typo after looking at Table 1- this is a self-efficacy question.

• Are business school graduates studying “management science”?

• The claim that low media visibility of women makes academic carers less attractive to them than for men is incorrect because the experiment would suggest the attractiveness of a career for women is not dependent on representation (but it is for men).

• I’m not sure it’s correct to say that other-gender role models can be counterproductive for men. Perhaps men have inflated self-efficacy and this helps calibrate it?

Section 5.1

• A newsletter should not be extrapolated to represent “media.” As such many claims in the discussion are too strong such as, “confirming that media’s representation of the world has negative performance effects not just on academia but on society in general.”

• Please temper all language in the discussion and do not generalize “management science” to science or STEM. Similarly, specify “newsletters” as opposed to “media.”

Section 5.3

• The first paragraph of the conclusion should be labeled limitations.

Reviewer #2: Thank you for the opportunity to review this work, I found it very interesting and worthwhile. This manuscript addresses a timely and socially significant issue: the impact of female researchers’ media visibility on public perceptions and academic career aspirations. The authors employ two well-structured experimental studies to explore how gender representation in media influences perceived expertise, self-efficacy beliefs, and the attractiveness of academic careers. The paper is clearly written, well-organized, and grounded in relevant theoretical frameworks, including social role theory.

The introduction is particularly comprehensive, offering a detailed review of the literature that effectively situates the study within broader academic and societal debates. However, there is an over-reliance on direct citations, which detracts from the authors’ own voice. Rephrasing some of these references in the authors’ own words would help clarify how each cited work supports the study’s rationale and hypotheses, and would better integrate the literature into the narrative. Additionally, the manuscript currently uses a mix of numerical and in-text citation styles, which should be standardized for consistency and clarity.

While the experimental design is methodologically sound and I appreciated the rationale behind the choice of design and stimuli, the manuscript would benefit from more cautious use of causal language. Phrases suggesting that media visibility “increases” self-efficacy or “erodes” perceptions may overstate the findings, particularly given the complexity of the social and psychological processes involved. A more measured phrasing—such as “is associated with” or “may influence”—would better reflect the limitations of experimental inference in this context.

Similarly, the formulation of Hypothesis 1 could be improved for clarity. The term “increase” is somewhat vague and would benefit from a more precise definition. It is not immediately clear whether the authors are referring to more positive perceptions, higher ratings on specific scales, or another outcome. As such, I recommend explicitly linking the hypothesis to the operationalized measures.

In a similar vein, the conclusions drawn in the discussion section should be more carefully quantified. While the study provides valuable insights into perceptions and self-reported beliefs, it does not measure actual career advancement or long-term outcomes, which the text sometimes alludes to. As such, claims about the broader impact on academic career trajectories should be moderated to reflect the scope of the data. Acknowledging this limitation more explicitly would strengthen the credibility of the conclusions.

Minor issues:

There are also some minor issues with redundancy and structure. For instance, the phrase “between-subjects laboratory experiments” is repeated unnecessarily on page 12 and could be streamlined. Furthermore, while the results of Study 1 are clearly presented, the manuscript would benefit from a brief discussion section following this study. Summarizing the key findings and their implications before transitioning to Study 2 would help maintain narrative coherence and reinforce the significance of the results.

In terms of statistical reporting, the manuscript currently reports p-values in threshold form (e.g., p < .05). Providing full p-values would enhance transparency and allow readers to better assess the strength of the evidence. Given the nuanced nature of the findings and the fact that one of the effects was not replicated across the 2 studies, I also encourage the authors to add effect sizes to their results.

In summary, this manuscript makes a valuable contribution to the literature on gender, media, and science communication. With revisions to improve clarity, consistency, and interpretive caution, it has strong potential for publication.

6. PLOS authors have the option to publish the peer review history of their article (what does this mean?). If published, this will include your full peer review and any attached files.

Reviewer #1: No

Reviewer #2: No

---

## [Author Response · Author response to Decision Letter 1]

29 Oct 2025

When Matilda shows up:

The double-edged impact of women researchers’ media visibility

(Manuscript Number: PONE-D-25-25301)

General comment

Dear Editors,

Dear Reviewers,

We are grateful for your thoughtful and detailed feedback on our manuscript. We have revised the manuscript extensively in response to your comments and suggestions. Below, we provide a detailed, point-by-point response. Each reviewer’s comment is reproduced in full, followed by our reply and a description of the changes made to the manuscript.

* * *

Detailed responses to Reviewer 1

Abstract

The first sentence uses ‘women,’ which indicates gender and ‘female’ which indicates sex. Please investigate how these terms are different and consider which is more appropriate for this study. I think that using terms that reflect gender is most important. People who appear as women have an impact on others who also identify as women.

Reply: We agree and have revised all uses of “females” and “males” to “women” and “men” in the abstract and throughout the manuscript where gender (not sex) is the relevant concept.

It is unclear whether the gains (increased perceived expertise and decrease the gender gap for self-efficacy beliefs) is specific to the public, grads, or both. Please include null results in the abstract as well.

Reply: We clarified in the abstract which effects apply to which populations and have added a sentence to acknowledge the non-significant findings. This is how it now appears:

Although the proportion of women in academic positions has increased, women researchers continue to receive less media coverage than their counterparts. To examine the implications of this underrepresentation, we investigate how non-academic audiences respond to women researchers’ increased visibility in science communication media. We conducted two laboratory experiments, the first with a sample of non-academic individuals (N = 271) and the second with a sample of graduate students (N = 129). Results show that greater visibility of women researchers reduces the gender gap in self-efficacy beliefs between men and women in both populations. It also increases women researchers’ perceived expertise among non-academic participants, but not among graduate students. Among the later, however, heightened visibility is also unexpectedly associated with a decline in the perceived attractiveness of academic careers, but only for men. While our research focuses specifically on the field of management and measures media visibility through the relative representation of women and men in a simulated newsletter, our findings provide broader insights into how media exposure can shape self-efficacy beliefs, researchers’ perceived expertise, and the appeal of academic careers. Building on these insights, we propose recommendations for researchers, academic institutions, and media organizations involved in science dissemination.

Temper language to ensure that “management science” is not being generalized to “science” and that a “newsletter” is not being generalized to all “media.”

Reply: Language has been revised in the abstract and throughout the manuscript to reflect the specific disciplinary (management science) and medium-specific (newsletter) nature of our study. This is how it now appears in the abstract:

Although the proportion of women in academic positions has increased, women researchers continue to receive less media coverage than their counterparts. To examine the implications of this underrepresentation, we investigate how non-academic audiences respond to women researchers’ increased visibility in science communication media. We conducted two laboratory experiments, the first with a sample of non-academic individuals (N = 271) and the second with a sample of graduate students (N = 129). Results show that greater visibility of women researchers reduces the gender gap in self-efficacy beliefs between men and women in both populations. It also increases women researchers’ perceived expertise among non-academic participants, but not among graduate students. Among the later, however, heightened visibility is also unexpectedly associated with a decline in the perceived attractiveness of academic careers, but only for men. While our research focuses specifically on the field of management and measures media visibility through the relative representation of women and men in a simulated newsletter, our findings provide broader insights into how media exposure can shape self-efficacy beliefs, researchers’ perceived expertise, and the appeal of academic careers. Building on these insights, we propose recommendations for researchers, academic institutions, and media organizations involved in science dissemination.

Introduction

In section 1 it seems odd that the impact on an academic audience (graduate students) is not mentioned here given the sample focused on that population.

Reply: We have updated the introduction in several ways following your comment. First, we acknowledge that graduate students could be considered an academic audience and have changed the title of the article from “When Matilda shows up: The double-edged impact of women researchers’ media visibility on a non-academic audience” to a more neutral “When Matilda shows up: The double-edged impact of women researchers’ media visibility”.

Second, your feedback made us realize that we had not sufficiently addressed the question of the target audiences for scientific dissemination activities in the introduction and the rationale for focusing on the general public and on graduate students in our research. We have therefore clarified these aspects as follows:

Despite the general rise in academic visibility, research dissemination in newspapers, radio and television, news websites, and dedicated online platforms seems to make little room for women researchers. Men also communicate more than women on social media [9] and personal blogs [13]. Women have made up some ground recently, but only 24% of expert voices heard in the news in 2020 were female (vs. 19% in 2015 ). Women researchers’ underrepresentation in the media is considered an enduring trend [4] with significant negative consequences. The extant research shows that this underrepresentation is detrimental to women academics’ careers [14,15] but has not explored its impact beyond peers from the same discipline and academic institutions. Yet scientific dissemination efforts are increasingly directed toward broader lay audiences [6,9], including the general public, to foster scientific literacy and engagement, the educational community, to integrate research findings into teaching and training and decision-makers and socio-economic stakeholders, to inform policies and professional practices. In this article, we specifically examine how the media visibility of women researchers influences two key audiences: the general public, as the primary target of scientific dissemination activities, and graduate students, who may be closer to the academic realm but for whom role models are particularly important in envisioning an academic career.

Third, following your next comment, we removed the summary of our findings from the end of the introduction and instead expanded the description of our empirical approach to clarify that we conducted two distinct experiments: one with a sample from the general public and the other with a sample of graduate students. This section now reads as follows:

In this article, we explore whether the visibility of women researchers in scientific dissemination media could enhance audience’s perceptions of women researchers’ expertise, self-efficacy beliefs, and the attractiveness of an academic career. To address this question, we conducted two laboratory experiments: the first with a sample of non-academic individuals (N = 271), representing the primary target of scientific dissemination activities, and the second with a sample of graduate students (N = 129), who may be closer to the academic sphere but for whom role models are particularly important in envisioning an academic career.

The introduction should be framed with this experiment in mind. Currently, these experiments are largely absent from the intro.

Reply: As you can see above, we have added further details to present the two experiments from the outset in the introduction.

End of section 1: It seems a bit unusual to summarize the findings here.

Reply: We have removed the summary from the end of Section 1 and repositioned it more appropriately in the Discussion section.

Section 2

Section 2.2. The word “refusing” seems strong here and perhaps an overstatement. “Decline” may be more appropriate.

Reply: We have replaced “refusing” with “declining,” as suggested.

Section 2.3.1 The sentence that says that lower media visibility weakens careers, salaries, self-esteem and wellbeing is a bit too strong. I suggest rephrasing by adding “can” weaken.

Reply: We have softened this claim by stating that lower media visibility “can weaken” these outcomes.

Section 2.3.3: It is unclear what is meant by counter-stereotypical women in research. This implies that the woman herself is counterstreotypic.

Reply: We have rephrased to refer to “counter-stereotypical portrayals of women,” making clear that we refer to representation, not individual identity.

Section 2.3.3: Paragraph after H1 (pg 10) sentence in the middle of a paragraph abruptly cuts off.

Reply: We have identified and corrected the incomplete sentence.

Section 3

Section 3.1

What is management science? Is this considered STEM? As a reviewer in the US, I was unfamiliar with this field and think many readers would appreciate a brief explanation and also what distinguishes it as a STEM field.

Reply: Thank you for raising this point. To avoid confusion and better reflect the scope of our stimuli, we have replaced the term “management science” with “management” throughout the manuscript. This broader term aligns more closely with the range of topics included in the newsletter stimuli and avoids suggesting a strict classification as a STEM field.

Typo near citation 121

Reply: Corrected.

How were graduate students recruited specifically? What language was used in the recruitment for the study? How were participants incentivized?

Reply: We expanded the Method section to include the missing information as follows:

We recruited 148 business school graduate students to participate in Study 2. Participants were contacted through HEC Montréal’s graduate student mailing list, completed the survey voluntarily without any incentive, and responded in French. Study 2 followed the same protocol as Study 1, with two notable exceptions.

For the sake of symmetry, we specified for Study 1 that the sample was composed of French participants:

We recruited 347 French participants through the online panel of a European professional market research institute.

In what ways were the participants told to interact with this newsletter? Presumably no lay person is going to randomly interact with a newsletter about management science. As such, how do you view this as an exercise that reflects lay people’s interaction with the media more broadly?

Reply: We now explain that participants were instructed to read the newsletter content carefully as part of the experimental design. We also clarify that our intervention simulates structured rather than organic exposure. In addition, we specify the online platforms in France that inspired our design, one of which has over 116k followers on YouTube. This indicates that lay audiences are indeed interested in engaging with online research dissemination newsletters that they can subscribe to in order to receive the latest updates.

The design of our experimental stimuli was inspired by actual newsletters distributed by online platforms that aim to popularize academic research (e.g., Xerfi Canal on YouTube, which displays more than 116k followers as of August 12, 2025). To maintain ecological validity, the men-dominated condition, featuring eight men researchers out of nine, mirrors real patterns of research dissemination observed in one of the authors’ home countries […]

To simulate deliberate rather than incidental media exposure, we explicitly instructed participants to read the newsletter content carefully, as they would be asked questions about its content afterward. We acknowledge that this structured exposure differs from organic media consumption; however, it is common and necessary in experimental settings where stimulus control is essential. After exposing participants to our stimuli, we measured their general self-efficacy beliefs. We then showed them a presentation of a research project being undertaken by “Clémence Dubois”, a fictitious woman researcher. It contained a short description of her fictitious project and a picture generated by the same AI used for the initial stimuli (see Figure 3).

The paper should acknowledge the presence of gender nonbinary and gender queer individuals. The paper can acknowledge their existence in science while still focusing on gender binary categories. Relatedly, were participants given the option to identify as gender non-binary?

Reply: We now detail in the Method section how gender was measured. We also report the number of participants who did not identify as a man or a woman and explain that they were excluded from the analyses due to the very small subgroup size: 3 participants in Study 1 (2 who described themselves as “human beings” and 1 who did not provide an answer) and 3 participants in Study 2 (1 who self-identified as “gay” and 2 as “non-binary”). This is how it now appears for Study 1:

Towards the end of the survey, participants were asked to indicate their gender (“woman,” “man,” “non-binary,” “prefer to self-describe [free-text field] ,” or “prefer not to say”), age, and highest educational qualification. […] We excluded from the analyses 3 participants who did not identify as either a man or a woman in order to focus on gender binary categories and 73 participants who provided an incorrect response regarding the gender composition of the newsletter.

I strongly disagree that “To what extent do you feel confident in your ability to express your opinion on a management issue” is reflective of perceived attractiveness of an academic career. Is this a previously validated scale to measure attractiveness of an academic career? I think this is a typo after looking at Table 1—this is a self-efficacy question.

Reply: You are absolutely right. This was a mistake. This item was a self-efficacy item indeed. We corrected it to properly show a perceived attractiveness of an academic career item : “To what extent do you feel academia would be a good work environment for you?”

Are business school graduates studying “management science”?

Reply: We replaced management science with management throughout the manuscript.

The claim that low media visibility of women makes academic carers less attractive to them than for men is incorrect because the experiment would suggest the attractiveness of a career for women is not dependent on representation (but it is for men).

Reply: You are right. We have modified our claim accordingly:

Specifically, women researchers’ lower media visibility has two critical impacts. It can weaken young women’s self-efficacy beliefs before they have even embarked on an academic career and may lead them to feel less legitimate in aspiring to such a career. It may then limit women researchers’ advancement once they have embarked on an academic career, by driving a perception that they are generally less expert than their counterparts.

I’m not sure it’s correct to say that other-gender role models can be counterproductive for men. Perhaps men have inflated self-efficacy and this helps calibrate it?

Reply: We have eliminated this idea from the discussion.

Section 5

Section 5.1. A newsletter should not be extrapolated to represent “media.” As such many claims in the discussion are too strong such as, “confirming that media’s representation of the world has negative performance effects not just on academia but on society in general.”

Reply: Thank you for pointing this out. We have revised Section 5.1 to more accurately describe the scope of our findings. We no

---

## [Decision Letter · Decision Letter 1]

30 Dec 2025

PONE-D-25-25301R1When Matilda shows up: The double-edged impact of women researchers’ media visibilityPLOS One

Dear Dr. Agogué,

Thank you for submitting your manuscript to PLOS ONE. After careful consideration, we feel that it has merit but does not fully meet PLOS ONE’s publication criteria as it currently stands. Therefore, we invite you to submit a revised version of the manuscript that addresses the points raised during the review process.

We look forward to receiving your revised manuscript.

Kind regards,

Robin Haunschild

Academic Editor

PLOS One

Journal Requirements:

Reviewers' comments:

Reviewer's Responses to Questions

**Comments to the Author**

1. If the authors have adequately addressed your comments raised in a previous round of review and you feel that this manuscript is now acceptable for publication, you may indicate that here to bypass the “Comments to the Author” section, enter your conflict of interest statement in the “Confidential to Editor” section, and submit your "Accept" recommendation.

Reviewer #1: (No Response)

Reviewer #2: All comments have been addressed

2. Is the manuscript technically sound, and do the data support the conclusions?

Reviewer #1: No

Reviewer #2: Yes

3. Has the statistical analysis been performed appropriately and rigorously? 

Reviewer #1: Yes

Reviewer #2: Yes

4. Have the authors made all data underlying the findings in their manuscript fully available?

Reviewer #1: (No Response)

Reviewer #2: Yes

5. Is the manuscript presented in an intelligible fashion and written in standard English?

Reviewer #1: Yes

Reviewer #2: Yes

6. Review Comments to the Author

Reviewer #1: I thank the authors for their revised work. I have two major concern that needs to be more thoroughly addressed before this manuscript should be considered for publication.

(1) Management is not a natural science and yet much of the introduction draws from literature specific to the underrepresentation of women in science, STEM, and science communication (which often exclusively is specific to the natural sciences). Management is treated by the European Research Council within Social Sciences & Humanities. Similarly, the OECD/UNESCO FOS system classifies business and management as a social science. As such, the paper needs to be fully revised to reflect this important classification.

For example, the title should specify “women researchers’ media visibility in the field of management.” Second, the term “scientific dissemination media” should be revised throughout to specify social science dissemination.

Further, the introduction should be revised to focus only the representation of women in the social sciences or preferably business/management research and exclude STEM and the natural sciences. Many of the references the authors make, including many of the papers cited in Table 1, are specific to STEM or the natural sciences, which is extremely misleading for the reader.

(2) The authors continue to conflate the non-academic audience and graduate student audience. The paper needs to be fully revised to reflect that this set of experiments examines the impact on social science researcher media visibility on two different audiences, one of which is non-academic and the other which is actively working within an academic system.

Until these revisions are made, I cannot fully evaluate the appropriateness of the manuscript for publication- however I note some small suggestions below.

Methods

Please specify what the online panel of European market research institute is. Please provide descriptive demographics (e.g., gender, race/ethnicity, college generation status) of the participants for study I and study II.

For Study 2 methods, please reiterate that HEC is a business school.

Please be more specific in your language (e.g., page 18- suggesting weaker gender-science stereotypes among people who are more familiar with the academic world.). First, the reader does not know what gender stereotypes exist in management research specifically. And, I think the more important difference between the non-academic audience and the graduate student audience may be that the grad audience is familiar with the “academic world of management” and not the “academic world” broadly.

Reviewer #2: (No Response)

7. PLOS authors have the option to publish the peer review history of their article (what does this mean?). If published, this will include your full peer review and any attached files.

Reviewer #1: No

Reviewer #2: No

---

## [Author Response · Author response to Decision Letter 2]

13 Feb 2026

Before addressing each comment in detail, we would like to express our sincere appreciation to both reviewers for their thoughtful and constructive feedback. We are particularly grateful to Reviewer 2, who indicated that our responses sufficiently addressed heir concerns and that the manuscript could be accepted in its current form. We carefully considered all remaining suggestions from Reviewer 1 and revised the manuscript accordingly, as detailed below.

Reviewer #1 Comment 1 : Management is not a natural science and yet much of the introduction draws from literature specific to the underrepresentation of women in science, STEM, and science communication (which often exclusively is specific to the natural sciences). Management is treated by the European Research Council within Social Sciences & Humanities. Similarly, the OECD/UNESCO FOS system classifies business and management as a social science. As such, the paper needs to be fully revised to reflect this important classification.

Before addressing Reviewer 1’s comments point by point, we would like to note that fully restructuring the manuscript is challenging at this stage, particularly as Reviewer 2 indicated that the paper could be accepted in its current form. Nonetheless, we carefully reflected on Reviewer 1’s suggestions and made substantial revisions across multiple sections of the manuscript. Specifically:

- We changed the title to explicitly specify the disciplinary scope by adding “management science”;

- We modified the introduction to more clearly establish the cross-disciplinary relevance of the phenomenon and to justify our empirical focus on the social sciences, particularly management science;

- We extended the literature review to highlight that many studies reviewed in Table 1 (marked with an asterisk) draw on social science data, showing that research on women’s lower visibility is not limited to STEM fields but also occurs in disciplines perceived as more gender-balanced, thereby reinforcing the cross-domain relevance of gendered visibility dynamics;

- We now explicitly specified “social science dissemination media” at key anchor points;

- We extended the discussion section, to include a limitation regarding the fact that the experiments only considered the field of management science.

These changes are described in greater detail in our responses below.

Reviewer #1 Comment 4 : Further, the introduction should be revised to focus only the representation of women in the social sciences or preferably business/management research and exclude STEM and the natural sciences. Many of the references the authors make, including many of the papers cited in Table 1, are specific to STEM or the natural sciences, which is extremely misleading for the reader.

We appreciate your emphasis on positioning management within the social sciences, and we fully agree with this classification. We have clarified this explicitly in the introduction.

At the same time, we would like to emphasize that the literature on gender bias in researchers’ visibility has developed largely in a cross-disciplinary manner rather than being confined to specific scientific domains. Among the 42 articles summarized in Table 1, approximately 45% explicitly include the social sciences, either as a primary focus or within a cross-disciplinary approach, while the remaining studies focus on STEM or life sciences. This reflects the prevailing approach in this research stream, in which shared mechanisms, such as gender stereotypes, role modeling, cumulative advantage, and institutional bias, are theorized as operating across academic fields rather than being domain specific.

Importantly, these theoretical frameworks are not rooted in the natural sciences but are widely applied in social science research on inequality and organizations. There is therefore little reason, based on the existing literature, to assume that mechanisms identified in one scientific domain would not operate in another.

Moreover, from a substantive perspective, examining these dynamics within management constitutes a contribution rather than a limitation. Gender bias is often presumed to be more salient in STEM and life sciences (e.g., Leslie et al., 2015; Banchefsky & Park, 2018). Demonstrating that similar processes shape media visibility in management and the social sciences, where gender-science stereotypes are commonly assumed to be weaker, thus provides particularly informative insights.

We thank you for encouraging us to clarify these points, which helped us improve the precision of the manuscript. Accordingly, we made the following changes.

First, we revised the introduction to position prior work as cross-disciplinary and to clarify that the present study extends these insights specifically to the field of management. We also added a dedicated paragraph to justify the relevance of management as the focal discipline and to articulate its ecological validity for the present research. This paragraph now reads as follows:

Building on these perspectives, we posit that women researchers’ lower visibility in scientific dissemination activities has the potential to reshape social representations of both research and researchers. In the present work, we investigate this phenomenon in the social sciences, focusing specifically on management science. We do so for two reasons. First, management science increasingly engages in research dissemination aimed at broad audiences and plays a central role in shaping organizational practices, leadership norms, and public discourse, making visibility dynamics in this field particularly consequential. Second, studying these dynamics in management science provides a conservative test: gender-science stereotypes are typically presumed to be weaker in the social sciences than in STEM and life sciences (e.g., Leslie et al, 2015; Banchefsky & Park, 2018). Demonstrating comparable visibility effects in this context therefore offers especially compelling evidence of the pervasiveness of gendered evaluative processes. Accordingly, we argue that these effects are likely to extend across all academic domains, from STEM to the social sciences.

Second, we now indicate this in Table 1 by marking with an asterisk the studies that draw on social science data, either as a primary focus or within a cross-disciplinary approach, thereby clarifying that our literature review builds on prior research spanning multiple academic domains. We also added the following paragraph:

Importantly, a substantial number of the studies reviewed in Table 1 (i.e., references marked with an asterisk), draw on data from the social sciences. This suggests that lower visibility is not confined to STEM or life sciences, where gender bias is often assumed to be most pronounced, but is also documented in disciplines typically perceived as more gender-balanced. The persistence of these patterns in the social sciences reinforces the argument that gendered visibility dynamics reflect broader evaluative processes operating across academic domains.

Third, we extended the discussion section, to include a limitation regarding the fact that the experiments only considered the field of management science.

A further limitation is that we examined these dynamics exclusively within the field of management. Although this focus limits generalizability, it also constitutes a substantive contribution, as gender bias is often presumed to be more salient in STEM and life sciences. Showing that similar dynamics operate in management and the social sciences—where such stereotypes are assumed to be weaker—offers particularly informative insights. The experimental design could therefore be extended to other forms of media, such as podcasts, television programs, or popular science magazines, in order to test whether the observed effects hold across different dissemination channels. Likewise, applying a similar approach to other academic disciplines beyond management, such as natural sciences, engineering, medicine, or the arts, could reveal whether the dynamics we observe are discipline-specific or more generalizable.

Reviewer #1 Comment 2 : The title should specify “women researchers’ media visibility in the field of management.”

Thank you for this helpful suggestion. We agree that explicitly specifying the disciplinary context strengthens the clarity and positioning of the article. We therefore revised the title to: “When Matilda Shows Up: The Double-Edged Impact of Women Researchers’ Media Visibility in Management Science.” This revised title preserves the conceptual framing while clearly anchoring the study within the field of management, in line with both the experimental stimuli and the theoretical contributions of the paper.

Reviewer #1 Comment 3 : The term “scientific dissemination media” should be revised throughout to specify social science dissemination.

We agree that clarifying the disciplinary scope of the dissemination context strengthens the precision of the manuscript. Accordingly, we now explicitly specify “social science dissemination media” at key anchor points: in the abstract, at the first occurrence in the introduction, and at the first mentions in both the discussion and conclusion. To preserve readability and avoid unnecessary repetition, we subsequently retain the shorter term “scientific dissemination media,” as the social science context is clearly established. This approach ensures both conceptual clarity and stylistic fluidity.

Reviewer #1 Comment 5 : The authors continue to conflate the non-academic audience and graduate student audience. The paper needs to be fully revised to reflect that this set of experiments examines the impact on social science researcher media visibility on two different audiences, one of which is non-academic and the other which is actively working within an academic system.

We took your previous comment from the first round of review into account; however, an unintended reference to a “non-academic audience” remained in the abstract in Revision 1. We agree that this wording required correction and have now revised it accordingly.

Reviewer #1 Comment 6 : Please specify what the online panel of European market research institute is.

We are now mentioning PanelLabs as the online panel of European market research institute.

Reviewer #1 Comment 7 : Please provide descriptive demographics (e.g., gender, race/ethnicity, college generation status) of the participants for study I and study II.

We were already giving age and gender as demographics for both studies. We added level of study for Study I. We unfortunately don’t have race/ethnicity or college generation status. We agree that these could play a role in our study, and are now mentioning this in the limitations and further research.

Reviewer #1 Comment 8 : For Study 2 methods, please reiterate that HEC is a business school.

We had already indicated in the Method section that “we recruited 148 business school graduate students to participate in Study 2. Participants were contacted through HEC Montréal’s graduate student mailing list.” However, we acknowledge that this may not have been sufficiently explicit. We therefore revised this passage to: “We recruited 148 graduate students enrolled in a business school (HEC Montréal) to participate in Study 2. Participants were contacted through the school’s graduate student mailing list, completed the survey voluntarily without any incentive, and responded in French.” We hope this revised wording provides greater clarity.

Reviewer #1 Comment 9 : Please be more specific in your language (e.g., page 18- suggesting weaker gender-science stereotypes among people who are more familiar with the academic world). First, the reader does not know what gender stereotypes exist in management research specifically. And, I think the more important difference between the non-academic audience and the graduate student audience may be that the grad audience is familiar with the “academic world of management” and not the “academic world” broadly.

We fully agree that this sentence in the Method section was misleading. Nonetheless, we believe it is noteworthy that in Study 1, 73 participants failed the manipulation check (i.e., recalling whether they had seen men or women in the newsletter), whereas in Study 2 only one graduate student did so. We therefore added a brief footnote to acknowledge this difference, without speculating on its underlying causes.

In addition, we revised the manuscript throughout to clarify the “more familiar” argument, specifying that the graduate student sample is familiar with the academic world of management rather than with academia more broadly. We thank the reviewer for this helpful clarification.

---

## [Decision Letter · Decision Letter 2]

12 Apr 2026

When Matilda shows up: The double-edged impact of women researchers’ media visibility in management science

PONE-D-25-25301R2

Dear Dr. Agogué,

We’re pleased to inform you that your manuscript has been judged scientifically suitable for publication and will be formally accepted for publication once it meets all outstanding technical requirements.

Kind regards,

Robin Haunschild

Academic Editor

PLOS One

Additional Editor Comments (optional):

Reviewers' comments:

Reviewer's Responses to Questions

**Comments to the Author**

1. If the authors have adequately addressed your comments raised in a previous round of review and you feel that this manuscript is now acceptable for publication, you may indicate that here to bypass the “Comments to the Author” section, enter your conflict of interest statement in the “Confidential to Editor” section, and submit your "Accept" recommendation.

Reviewer #1: All comments have been addressed

2. Is the manuscript technically sound, and do the data support the conclusions?

Reviewer #1: Yes

3. Has the statistical analysis been performed appropriately and rigorously? 

Reviewer #1: Yes

4. Have the authors made all data underlying the findings in their manuscript fully available?

Reviewer #1: Yes

5. Is the manuscript presented in an intelligible fashion and written in standard English?

Reviewer #1: Yes

6. Review Comments to the Author

Reviewer #1: I appreciate the authors for specifying their language throughout the manuscript. All comments have been addressed.

7. PLOS authors have the option to publish the peer review history of their article (what does this mean?). If published, this will include your full peer review and any attached files.

Reviewer #1: No

---

## [Editor Report · Acceptance letter]

PONE-D-25-25301R2

PLOS One

Dear Dr. Agogué,

I'm pleased to inform you that your manuscript has been deemed suitable for publication in PLOS One. Congratulations! Your manuscript is now being handed over to our production team.

Kind regards,

on behalf of

Dr. Robin Haunschild

Academic Editor

PLOS One